# Prevalence of and Factors Associated with Negative Psychological Symptoms among Elderly Widows Living Alone in a Chinese Remote Sample: A Cross-Sectional Study

**DOI:** 10.3390/ijerph20010264

**Published:** 2022-12-24

**Authors:** Hui You, Yao Wang, Lily Dongxia Xiao, Li Liu

**Affiliations:** 1Department of Midwifery, Xiangya School of Nursing, Central South University, Changsha 410013, China; 2College of Nursing and Health Sciences, Flinders University, Adelaide, SA 5042, Australia; 3Department of Community Nursing, Xiangya School of Nursing, Central South University, Changsha 410013, China

**Keywords:** widowhood, aged, women, depression, anxiety, loneliness

## Abstract

(1) Background: Research indicates that most elderly widows are at a high risk of experiencing negative psychological symptoms. It is common for elderly women in rural and remote areas to live alone without family support to cope with stress due to the mass rural-to-urban migration of China’s youth labor force. Such a situation further worsens their psychological health and well-being. However, the prevalence of and risk factors associated with negative psychological symptoms (loneliness, depression, and anxiety) among remote elderly widows living alone in China are currently unclear; (2) Methods: A cross-sectional study was conducted in Hunan Province, China. The loneliness was assessed through the University of California at Los Angeles Loneliness Scale (ULS-8). The depression and anxiety were assessed with the Short Form Geriatric Depression Scale (GDS-15) and Generalized Anxiety Disorder Scale (GAD-7), respectively. The Chi-square test and correlation analysis were conducted to identify factors associated with negative psychological symptoms. Logistic regression was performed to predict risk and protective factors contributing to loneliness, depression, and anxiety symptoms. The significance level was set as *p* < 0.05; (3) Results: A total of 271 remote elderly widows living alone were enrolled in the present study. Additionally, 234 valid questionnaires were returned (valid response rate = 86.3%). The prevalence of loneliness, depression and anxiety was 8.1%, 44.0%, and 16.7%, respectively. Acute or chronic medical conditions, marital happiness, being the primary caregiver before widowhood and anticipating the death of the spouse differed significantly in the distribution of negative psychological symptoms. Logistic regression analysis predicted that participants who were satisfied with their marriage had a lower likelihood to experience loneliness, depression, and anxiety (*p* < 0.05). Being the spouse’s primary caregiver before widowhood was more likely to have symptoms of loneliness (*p* < 0.01). Those with various acute or chronic medical conditions were more likely to suffer from depression (*p* < 0.01); (4) Conclusions: Remote elderly widows living alone in China are prone to loneliness, depression, and anxiety symptoms. Being the primary caregiver before widowhood and having many acute or chronic medical conditions are risk factors for loneliness and depression, respectively. Marital happiness is the protective factor against negative psychological symptoms. To accomplish the goal of equitable access to mental health care in China, evidence-based policy and resource development to support psycho-social interventions that prevent and manage negative psychological symptoms for remote elderly widows living alone are urgently needed.

## 1. Introduction

Like other nations in the world, China is experiencing a rapidly aging population and the population aged 65 and over has reached 13.50% in the latest census [1]. Chinese women have a longer average life expectancy (80.5) than Chinese men (74.7); hence, elderly women are significantly more likely to be widowed than elderly men [2,3]. In 2010, China’s sixth census revealed that of the 47.74 million elderly widows, 33.45 million were female. Such characteristics of female-dominant elderly widows are also reflected in reports from other countries [4,5]. It is estimated that the number of elderly widows in China will reach 94.49 million by 2050 [2]. Studies revealed that widowhood in older women is associated with poorer mental health and reduced coping resources compared with those living with their partners [6,7,8]. Older female widows living alone in rural and remote areas even face more stressors and fewer coping resources in their daily lives compared with their counterparts in urban areas [9,10,11]. However, studies on the prevalence of and factors associated with their psychological well-being for this most vulnerable population are scarce. This study addresses the gaps in the research. 

It is well-researched that elderly widows experience numerous changes including cognitive decline [12], sleep problems [13], and decreased life satisfaction [14]. While those changes gradually declined in the majority of elderly widows, some of them developed persistent and negative psychological symptoms such as loneliness, depression, and anxiety that can lead to increased morbidity and mortality rates among older widows [15,16,17,18], main factors affecting society to achieve the WHO and the UN healthy aging and quality of life for older people [19,20]. Loneliness is defined as a psychological condition resulting from the discrepancy between an individual’s existing and desired social relationships [21]. The prevalence of loneliness among elderly widows in China is 40.7% [22]. Studies show that nearly 70% of elderly widows find loneliness is the most difficult aspect of daily life to deal with [23]. Anxiety is defined as apprehensive anticipation of future danger or misfortune, accompanied by feelings of unease or physical symptoms of tension [24]. Depression is often characterized by persistent sadness and a lack of interest or pleasure in previously valuable or enjoyable activities [25]. The prevalence of depression is about 10–45% in elderly widows worldwide [8,10,26]. 

The social production function theory developed by Lindenberg explains the likelihood of negative psychological symptoms in elderly widows living alone from a sociological perspective [27,28]. Lindenberg believes that people are goal-oriented and seek to produce psychological or emotional well-being [29]. A person’s psychological well-being is determined by physical and social well-being. The achievement of physical and social well-being depends on the achievement of instrumental goals, such as comfort to produce physical well-being and affection to produce social well-being. A lack of resources can prevent people from generating well-being and induce depressive symptoms. Being married and living with a spouse is a great resource for generating all the instrumental goals that contribute to well-being. On the contrary, widowhood and living alone are associated with a lack of resources, and affection to engage in meaningful physical and social activities; thus, contributing to poor well-being and negative psychological symptoms in this population.

Studies identified that long periods of widowhood, declining income, or managing multiple chronic illnesses in elderly widows were associated with these negative psychological symptoms [30,31]. However, the mental health of elderly widows attracted less attention in China, as evidenced by a relatively small number of studies [10,32,33,34]. Such a situation also indicates that many elderly widows remain undiagnosed and untreated in China. Remaining in this situation could increase the burden on China’s healthcare system and hinder the goal of achieving equitable access to healthcare.

In China, the huge disparities in economic development between urban and rural areas cause a large proportion of China’s young and middle-aged rural labor force to migrate to urban areas for employment opportunities, while most rural elderly widows live alone [35]. Such internal migration is more prevalent in remote areas where employment opportunities are even less compared with rural areas [36,37]. Chinese remote elderly female widows living alone may experience multiple risk factors of poor health and well-being that are widely reported in the literature including older age [38,39], female gender [8,31,40], living arrangements [41], and rural and remote region [10,42]. However, few empirical studies have carefully investigated the prevalence of and factors associated with negative psychological symptoms among them. Therefore, a cross-sectional study was conducted in remote regions of southeastern China to determine the prevalence and risk factors of negative psychological symptoms, including loneliness, depression, and anxiety.

## 2. Materials and Methods

### 2.1. Ethics

This study has been reviewed and approved by the Ethics Committee of Xiangya School of Nursing, Central South University (Project Number 2019021). Participants finished the questionnaires anonymously. The questionnaires were then coded uniformly by the investigators. Participants who scored above the cut-off value on the self-rating scales of loneliness, depression, and anxiety were offered psychological counseling by investigators within 48 hours. Investigators assisted with referrals to medically designated institutions or mental health professional institutions for those whose negative psychological symptoms did not improve after psychological counseling.

### 2.2. Study Design 

A cross-sectional study was conducted to investigate the loneliness, depression, and anxiety of remote elderly widows living alone in China. This study was conducted from December 2018 to March 2019. 

### 2.3. Study Setting

This study was conducted in Longshan County, Hunan Province, China. Longshan County is located on the northwestern edge of Hunan Province and is one of the most remote counties in Hunan. The per capita Gross Domestic Product (GDP) of Longshan County in 2021 (RMB 20, 409) is still below China’s per capita GDP (RMB 80, 976) and the per capita disposable income (RMB 17, 560) is also below the country’s average (RMB 35, 128) [43,44]. Longshan county has a relatively diverse population, with ethnic minorities accounting for 81% of the country’s total population [45]. The proportion of people aged 60 and over in Longshan County (18.99%) is similar to the national level (18.70%). In addition, the sex ratio (the ratio of males to females, using females as 100) of 104.66 for the population of Longshan County is also very close to that of the country (105.07) [46,47], predicting a similar proportion of remote elderly widows living alone as in China. Given the same social status and traditions of remote elderly widows living alone in China, the survey conducted in Longshan County should be representative.

### 2.4. Participants and Recruitment

Based on the list of elderly women registered with the village committee, and combined with recommendations from the committee and villagers, the investigators invited elderly widows to participate in the study if they met the inclusion and exclusion criteria. The inclusion criteria were as follows: (1) female ≥60 years old; (2) not remarried after widowhood; (3) living alone due to widowhood and not accompanied by children or other partners; (4) having clear awareness and communication skills; and (5) voluntary participation in the study. Individuals who met one of the following criteria were excluded: having a severe psychiatric illness, severe acute or non-communicable disease, or having undergone major surgery within the last year. 

In this study, the required sample size was calculated using the logistic regression formula from the textbook: $N=\frac{{\left(Z_{1-\alpha /2\;\;}+Z_{\beta }\right)}^{2}}{\left[P_{1}\left(1-P_{1}\right)\right]b^{2}}$, α = 0.05 [48]. *P*_1_ is the rate of positivity when x is taken as the mean value, and *b* is the estimated value of the regression coefficient corresponding to x. *P*_1_ and *b* were identified from the pilot study of participants. A sample size of 230 was required, taking into account a 15% missed visit rate. 

Trained investigators explained the purpose and process of the study to remote elderly widows living alone by telephone. They were informed that they were at liberty to decide whether to participate in the study without any consequences. Appointments were made for potential participants who had agreed to participate in the study for an in-home survey. After participants completed the informed consent form, investigators instructed them to complete the questionnaires. The contents of the questionnaires were verified in time after completion to eliminate invalid questionnaires.

### 2.5. Measures 

#### 2.5.1. Social-Demographic and Widowhood-Related Information

Social-demographic data, including age, race, education, occupation, acute and chronic medical conditions, and the number of children, were collected through a self-designed questionnaire. Widowhood-related information including the duration of widowhood, age at death of the spouse, marital happiness, living alone period, cause of death of the spouse, length of the marriage, being the primary caregiver of the spouse before widowhood, and anticipation of the death of the spouse was also collected.

#### 2.5.2. The University of California at Los Angeles Loneliness Scale (ULS-8)

The ULS-8 was used to evaluate the loneliness of the participants [49]. The scale consists of eight items and uses a four-point Likert scale: 1 “never”, 2 “rarely”, 3 “sometimes”, and 4 “always”. The total score of the ULS-8 ranged from 8 to 32. The higher scores, the higher levels of loneliness. The Chinese version of the ULS-8 demonstrated good reliability and validity. The Cronbach’s α coefficient is 0.741. Exploratory factor analysis showed that the cumulative variance contribution was 55.39% when extracting one common factor from the Chinese version of the ULS-8 scale, which was consistent with the theoretical conception of the original scale [50]. The cut-off value for ULS-8 in this study used the 24 points determined in previous studies [51,52].

#### 2.5.3. Short-Form Geriatric Depression Scale (GDS-15)

The GDS-15 was used to assess participants’ depressive symptoms during the last week. It includes 15 items. The response options of “yes” or “no” are adopted. Items 1, 5, 7, 11, and 13 were negatively worded and scored in reverse in a certain order. Higher scores indicate more severe depressive symptoms [53]. The Chinese version of the GDS-15 has good reliability, with a Cronbach’s alpha coefficient of 0.793. The Activity of Daily Living Scale assessed significant differences in GDS-15 scores among elderly people with different self-care abilities, demonstrating the GDS-15 scale’s good discriminant validity. The cut-off value for the Chinese version of the GDS-15 was 5 [54].

#### 2.5.4. Generalized Anxiety Disorder Scale (GAD-7)

Participants’ anxiety in the past two weeks was measured by the GAD-7. It consists of seven items whose scores range from 0 “not at all” to 3 “nearly every day”. A higher total score represents the more serious anxiety symptoms [55]. The cut-off value of the Chinese version of the GAD-7 scale score was 9. The Cronbach’s α coefficient of the scale was 0.898. The criterion validity of the Chinese scale was high, with a sensitivity of 86.2%, a specificity of 95.5%, and a kappa value of 0.825 [56].

### 2.6. Data Analyses 

Epidata 3.1 (The EpiData Association, Odense, Denmark) was used for data entry. The data were analyzed by SPSS 26.0 (Statistical Product and Service Solutions, IBM Corp., Armonk, NY, USA). All measurement data were assessed for normal distribution using the Kolmogorov–Smirnov test. The composition ratio, mean and standard deviation are used for statistical description. A chi-square test and correlation analysis were used in univariate analysis to test for differences in loneliness, depression, and anxiety symptoms across socio-demographic factors and widowhood-related factors. Binary logistic regression was used to predict factors contributing to the three negative psychological symptoms. The significance level was set as *p* < 0.05.

## 3. Results

### 3.1. Socio-Demographic Characteristics of Remote Elderly Widows Living Alone

A total of 271 remote elderly widows living alone were enrolled in the present study. Additionally, 234 valid questionnaires were returned (valid response rate = 86.3%). The mean age of the participants was 73.29 ± 7.37. The majority (69.7%) of participants had acute or chronic medical conditions. The vast majority (97.3%) of participants had been married for more than ten years. Participants who had been widowed for more than three years amounted to 78.7%. Almost eighty percent (80.8%) of widows were satisfied with their marriage. Approximately 60.3% of the participants were their spouse’s primary caregivers before they were widowed (Table 1). 

### 3.2. Prevalence of Loneliness, Depression, and Anxiety

The prevalence of loneliness was 8.1%. The mean ULS-8 score for the lonely participants was 24.68 ± 1.00. Almost half (44.0%) of widows had depression symptoms. Participants’ mean score of GDS-15 was greater than the cut-off value (5.29 ± 3.88). The mean GAD-7 score was relatively low (4.50 ± 4.10), and the proportion of widows with anxiety was 16.7% (Table 2). 

### 3.3. Comorbidity between Loneliness, Depression, and Anxiety 

Loneliness and depression are comorbid in 6.0% of the participants. Anxiety and loneliness have a 3.8% comorbidity. Additionally, 15.0% of participants experienced anxiety and depression. There are 3.8% of participants with all three of these negative psychological symptoms (Figure 1).

### 3.4. Associated Factors with Loneliness, Depression, and Anxiety

Univariate analyses were conducted to test the differences in socio-demographic and widowhood-related factors in loneliness, depression, and anxiety symptoms. Acute or chronic medical conditions, marital happiness, being the primary caregiver before widowhood and anticipating the death of the spouse were associated with negative psychological symptoms (*p* < 0.05) (Table 3).

Loneliness, depression, and anxiety symptoms were each examined as the dependent variable in a binary logistic regression analysis. The univariate analysis’s significant variables served as the independent variables. Marital happiness (OR = 0.283, *p* < 0.05) was found to be the protective factor against loneliness. Being the primary caregiver of the spouse before widowhood was a risk factor for loneliness (OR = 7.101, *p* < 0.01). Marital happiness was also the protective factor for depression and anxiety (OR = 0.332, *p* < 0.01; OR = 0.194, *p* < 0.001, respectively). Acute or chronic medical conditions are a risk factor for depression (OR = 2.257, *p* < 0.01) (Table 4).

## 4. Discussion

Remote elderly widows living alone in China are a vulnerable population. However, there are few comprehensive assessments of this population’s loneliness, depression, and anxiety prevalence. In this study, the prevalence rates for these negative psychological symptoms were 8.1%, 44.0%, and 16.7%, respectively. Furthermore, comorbidities of these symptoms are frequently present. The high prevalence of loneliness, depression and anxiety among remote elderly widows living alone suggests that this population has substantial mental health care needs.

Based on previous research, elderly widows living alone face numerous challenges and are prone to negative psychological symptoms [8,41]. Remote elderly widows living alone lack a supportive environment and adequate resources that can help maintain their physical and mental capacities [57,58,59]. The prevalent negative psychological symptoms lead to a decrease in the quality of life of elderly widows and an increase in the risk of hospitalization, which is detrimental to global healthy aging [19]. 

In China’s traditional family-oriented culture, the elderly place a high value on the daily companionship of their spouse [60]. As a result, elderly widows in China experience greater symptoms of loneliness than widows in other countries [61]. Besides, in remote areas, the willingness of elderly widows to remarry is weak due to the constraints of children’s opposition, public opinion, and traditional attitudes [62,63]. Few elderly widows had remarried, which may also contribute to the high prevalence of loneliness among remote elderly widows living alone. Local government officials should positively encourage remarriage by shaping public opinion to activate the spouse’s role in providing moral support. Additionally, other support systems should be bolstered to improve spiritual companionship. The government should encourage children to provide financial support or care to elderly widows and foster a harmonious and supportive neighborhood. 

As a result of providing care for their partners, older women are more prone to suffering from poor mental health [30]. After the death of their spouse, they have far less daily contact and communication with their intimate partner, which increases their risk for mental health issues [64,65]. The caregiver role in this study predisposed the experience of loneliness, contrary to the results of earlier research [29]. Our finding indicates that the relationship between the pre-widowhood caregiver role and negative psychological symptoms is inconclusive and requires further research. 

The participants’ pervasive depression and anxiety may result from living alone, having no one to discuss their negative feelings about widowhood, and the lack of recreational facilities and activities in remote areas [66]. Additionally, remote elderly widows always neglect their mental health and lack the knowledge to utilize the social security system to improve their physical and psychological health [67]. Village Councils need to strengthen social support for this vulnerable group by engaging them in self-care activities, for instance, regular seminars on mental health, self-assessment of mental health, and seeking health from health professionals for negative psychological symptoms. Secondly, healthcare workers in remote rural areas should receive relevant training to enhance their ability to identify and treat negative psychological symptoms in elderly widows. They also need to frequently visit elderly widows at risk of developing negative psychological symptoms, for instance living alone and having acute or chronic diseases, and encourage them to actively seek help from family members and close friends.

Participants who have more acute or chronic medical conditions are suspectable to depression. Therefore, primary care health professionals need to monitor the mental health conditions of this group, take preventive measures and provide timely interventions to this group of elderly windows. Regarding medical care, regular free medical checkups for remote elderly widows living alone should be added to public health service programs to reduce the risk of negative psychological symptoms in this population. The types of illnesses covered by medical insurance for elderly widows living alone should also be expanded by covering mental health problems. 

We found that marital happiness was a protector for elderly widows from developing negative psychological symptoms in this study. The finding is similar to previous studies which identified that marriage could provide the elderly with emotional satisfaction and physical health benefits [29,68,69]. Nonetheless, widowhood dissolves the marriage and deprives the spiritual support of the spouse, resulting in negative psychological symptoms [70,71,72]. One possible factor contributing to our finding might be that the vast majority had been widowed over 3 years. Therefore, they might have adapted to a new life without their partners. We suggest that future studies need to add a qualitative study component to enhance an understanding of the influence of marital happiness on the adaptation to widowhood in the elderly.

Research evidence strongly suggests that negative psychological symptoms were due to the difficulty of adapting to the change in widowhood (e.g., the decline in income, coping with the chores of life alone) after being married for a relatively extended period [30,73,74]. Village Councils should assist elderly widows in maintaining contact with friends or family members and meeting their spiritual and material needs. Moreover, early interventions should be developed to aid them in adjusting to widowhood sooner. Group-based complicated grief therapy and mindfulness techniques [75,76], as well as individual writing-based emotional expression therapy [77,78], are effective in adapting to widowhood. These evidence-based interventions should be considered as strategies to prevent remote elderly widows who live alone from developing mental health problems. 

Szczepanska-Gieracha and colleagues identified that older women’s sense of responsibility for their health through goal-focused group psychotherapy subsequently improved their depression symptoms [79]. Their studies provided new sight into the treatment of negative psychological symptoms in elderly widows living alone. The goal-focused group psychotherapy aims to establish group relationships among older women in similar situations and encourages them to discharge their negative feelings. The therapists then help to model older women’s right attitudes and motivate them to become more independent in their lives. This kind of psychotherapy can significantly reduce depressive symptoms in elderly women and improve their sense of well-being. Moreover, Szczepanska-Gieracha and colleagues also identified that the Virtual Therapy Garden provided for older women helped them better recognize and use relevant mental health resources [80]. This kind of Virtual Therapy includes stories and interactions in the Virtual Therapy Garden by which older women realized that there is much they can do to improve their mental health and take responsibility for their future. Virtual Therapy demonstrates a significant reduction in anxiety and depression. These evidence-based interventions should be considered as strategies to improve the mental health problems of remote elderly widows who live alone. 

To our knowledge, this is the first study to examine depression, anxiety, and loneliness in remote elderly widows living alone. Most previous studies focused on elderly widows and explored only a portion of the three negative psychological symptoms [8,41,81]. Besides, participants in this study were well represented. Longshan County is a typical remote area with backward economic development [82], and the age and gender structure of remote elderly widows living alone are extremely similar to national levels.

Some limitations need to be acknowledged in the current study. Our research is a cross-sectional study, thus causal inferences about the correlation cannot be made. A longitudinal study should be considered in the future. Second, participants’ self-rated loneliness, depression, and anxiety may have been somewhat influenced by recall bias. Finally, the results of this study may only be generalized to areas with similar economies and cultures. Based on the poor mental health status of this population and the great health care needs, there is a request for future research on a wider population.

## 5. Conclusions

Our research revealed that remote elderly widows living alone in China frequently experience loneliness, depression, and anxiety. Marital happiness was the protective factor against three negative psychological symptoms. Elderly widows who have been the primary caregiver for their spouses and suffering from acute or chronic illness are more likely to feel lonely or depressed. There is an immediate need for individualized psychosocial intervention programs and related services to relieve these symptoms and improve their life outcomes, contributing to the achievement of universal healthcare equality in China. 

## Figures and Tables

**Figure 1 ijerph-20-00264-f001:**
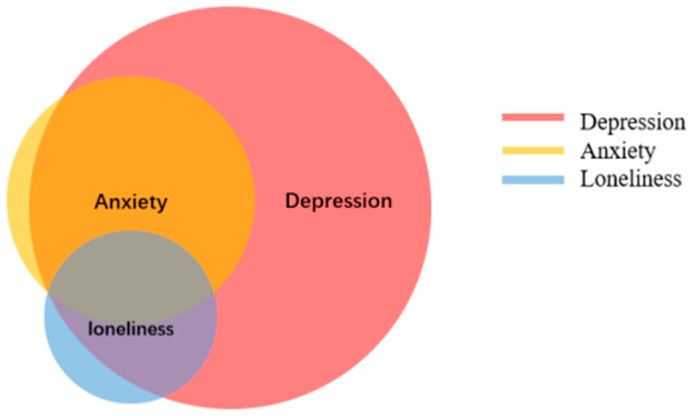
Comorbidity between loneliness, depression, and anxiety in widows (N = 234).

**Table 1 ijerph-20-00264-t001:** Demographic characteristics of the remote elderly widows living alone (N = 234, % or mean ± SD).

Variable	N (%)	Variable	N (%)
**Age**	73.29 ± 7.37	1~3	46 (19.7)
**Race**		3~10	96 (41.0)
Han	189 (80.8)	>10	75 (32.1)
Minorities	45 (19.2)	**Length of marriage (y)**	
**Education level**		<10	6 (2.7)
Primary school and below	212 (94.2)	≥10	216 (97.3)
Junior school	9 (4.0)	**Marital happiness**	
High/Specialized Secondary school or above	4 (1.8)	Yes	189 (80.8)
**Occupation**		No	45 (19.2)
Farmer	181 (83.8)	**Cause of death of the spouse**	
Housewife	25 (11.6)	Acute diseases	69 (29.5)
Employees of enterprises and institutions	7 (3.2)	Chronic diseases	113 (48.3)
Others	3 (1.4)	Accident	26 (11.1)
**Acute or chronic medical conditions**		Natural senescence	26 (11.1)
Yes	70 (30.3)	**As primary caregiver**	
No	161 (69.7)	Yes	93 (39.7)
**Number of children**		No	141 (60.3)
<3	128 (54.7)	**Age at death of the spouse (y)**	
≥3	106 (45.3)	< 45	22 (9.6)
**Duration of widowhood (y)**		45~59	53 (22.5)
<1	12 (5.1)	60-74	113 (48.3)
1~3	38 (16.2)	≥75	46 (19.6)
3~10	83 (35.5)	**Anticipate the death of the spouse**	
>10	101 (43.2)	Yes	116 (49.6)
**Living alone period (y)**		No	118 (50.4)
<1	17 (7.2)		

**Table 2 ijerph-20-00264-t002:** Descriptive statistics of widow’s loneliness, depression, and anxiety (N = 234, % or mean ± SD).

Variable	N (%)	Mean (SD)	Range
**Loneliness**		18.72 ± 3.54	8~27
Not-lonely	215 (91.9)	18.20 ± 3.18	
Lonely	19 (8.1)	24.68 ± 1.00	
**Depression**		5.29 ± 3.88	0~15
Non-Depressed	131 (56.0)	2.38 ± 1.56	
Depressed	103 (44.0)	9.00 ± 2.56	
**Anxiety**		4.50 ± 4.10	0~17
Non-anxious	195 (83.3)	3.07 ± 2.50	
Anxious	39 (16.7)	11.79 ± 2.68	

Lonely: ULS-8 score ≥24. Anxious, GAD-7 score ≥9, Depressed, GDS-15 score ≥5.

**Table 3 ijerph-20-00264-t003:** Univariate analysis of loneliness, depression, and anxiety (N = 234).

Variable	Loneliness		Depression		Anxiety	
	χ^2^/Fisher, Pearson’s r	*p*-Value	χ^2^/Fisher, Pearson’s r	*p*-Value	χ^2^/Fisher, Pearson’s r	*p*-Value
**Age**	0.042	0.522	0.038	0.563	0.084	0.200
**Race**	1.009	0.315	0.004	0.949	2.427	0.119
**Occupation**	1.802	0.615	3.476	0.324	5.782	0.123
**Education level**	0.402	0.818	3.000	0.221	0.388	1.000
**Acute or chronic medical conditions**	0.137	0.712	8.187	0.004 **	3.260	0.071
**Number of children**	0.449	0.503	0.937	0.333	0.345	0.557
**Duration of widowhood (y)**	0.040	0.543	0.012	0.850	0.029	0.661
**Living alone period (y)**	0.040	0.547	0.055	0.400	0.086	0.189
**Length of marriage (y)**	0.028	0.677	-0.036	0.591	-0.071	0.284
**Marital happiness**	4.129	0.042 *	11.599	0.001 **	21.840	<0.001 ***
**As primary caregiver**	4.955	0.026 *	0.082	0.775	0.032	0.858
**Age at death of the spouse (y)**	2.277	0.517	−0.012	0.851	−0.019	0.768
**Cause of death of the spouse**	1.432	0.231	1.405	0.236	1.508	0.219
**Anticipate the death of the spouse**	4.358	0.037 *	5.741	0.332	4.666	0.458

* χ2 test and Fisher’s exact test were used for the categorical variable. Pearson’s r was used for the continuous variable. * *p* < 0.05, ** *p* < 0.01, *** *p* < 0.001.

**Table 4 ijerph-20-00264-t004:** Binary logistic regression analysis of loneliness, depression, and anxiety (N = 234).

Variables	Model 1					Model 2					Model 3				
	B	SE	Exp (B)	95%CI	*p*-value	B	SE	Exp (B)	95%CI	*p*-value	B	SE	Exp (B)	95%CI	*p*-value
**Age**	1.147	0.788	3.148	0.671−14.758	0.146	0.190	0.411	1.209	0.540−2.707	0.644	0.596	0.527	1.815	0.646−5.105	0.258
**Race**	−0.858	0.857	0.424	0.079−2.272	0.317	−0.50	0.375	0.951	0.456−1.984	0.893	−0.920	0.605	0.398	0.122−1.303	0.128
**Occupation**	−1.536	1.059	0.215	0.027−1.716	0.147	0.256	0.336	1.292	0.669−2.495	0.446	0.144	0.462	1.155	0.467−2.854	0.755
**Education level**	1.092	1.157	2.979	0.309−28.758	0.345	−0.990	0.636	0.372	0.107−1.293	0.120	−1.103	1.133	0.332	0.036−3.057	0.330
**Acute or chronic medical conditions**	−0.120	0.600	0.887	0.274−2.873	0.841	**0.814**	**0.311**	**2.257**	**1.228−4.149**	**0.009** **	0.831	0.493	2.296	0.874−6.033	0.092
**Number of children**	−0.091	0.585	0.913	0.290−2.875	0.876	−0.159	0.315	0.853	0.460−1.581	0.614	−0.242	0.429	0.785	0.339−1.820	0.573
**Duration of widowhood (y)**	−0.209	0.545	0.812	0.279−2.362	0.702	−0.346	0.288	0.707	0.402−1.245	0.230	−0.629	0.404	0.533	0.241−1.176	0.119
**Living alone period (y)**	−0.116	0.331	0.890	0.465−1.702	0.725	0.210	0.172	1.233	0.880−1.729	0.224	0.279	0.256	1.322	0.801−2.182	0.275
**Length of marriage (y)**	0.742	0.902	2.101	0.359−12.298	0.410	−0.311	0.434	0.733	0.313−1.714	0.473	−0.784	0.579	0.456	0.147−1.419	0.175
**As primary caregiver**	**1.960**	**0.694**	**7.101**	**1.821–27.687**	**0.005** **	0.025	0.336	1.026	0.531−1.981	0.940	0.176	0.469	1.192	0.476−2.989	0.708
**Marital happiness**	**−1.263**	**0.543**	**0.283**	**0.098–0.819**	**0.020** *	**−1.104**	**0.355**	**0.332**	**0.165–0.665**	**0.002** **	**−1.638**	**0.383**	**0.194**	**0.092-0.412**	**<0.001** ***
**Age at death of the spouse (y)**	−0.571	0.489	0.565	0.217–1.473	0.243	0.171	0.247	1.186	0.731–1.925	0.490	0.280	0.337	1.323	0.684–2.560	0.405
**Cause of death of the spouse**	−0.322	0.359	0.725	0.358–1.466	0.370	−0.132	0.163	0.876	0.637–1.206	0.418	−0.258	0.220	0.773	0.502–1.188	0.240
**Anticipate the death of the spouse**	−0.988	0.541	0.372	0.129–1.076	0.068	−0.260	0.326	0.771	0.407–1.460	0.425	−0.499	0.456	0.607	0.248–1.483	0.273

95% CI: 95% Confidence Interval. Model 1: Loneliness as the dependent variable, Model 2: Depression as the dependent variable, Model 3: Anxiety as the dependent variable. * *p* < 0.05, ** *p* < 0.01, *** *p* < 0.001.

## Data Availability

Data for this study are available from the corresponding authors upon reasonable request.

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
