# Peer review of "Prevalence of and Factors Associated with Negative Psychological Symptoms among Elderly Widows Living Alone in a Chinese Remote Sample: A Cross-Sectional Study"

_ijerph, 2022, doi:10.3390/ijerph20010264_

Round 1

Reviewer 1 Report

My comments on " Prevalence of and Factors Associated with Negative Psychological Symptoms among Elderly Widows Living Alone in A Chinese Remote Sample: A Cross-sectional Study" (ijerph-2020947) are as follows.

 The aim of this research was to determine the prevalence and risk factors of negative psychological symptoms among remote elderly widows in China. To this aim, it has been prepared an cross-sectional survey which was attended by more than 230 participants.

 The topic of the manuscript is very interesting. I attempt to put forward only few suggestions to improve the manuscript.

-- First of all, the introduction is too simple. Then, you must do a literature review or a conceptual/ theoretical framework to better interpret current findings.

--Secondly, considering the final sample size is 234, some important confounders should be contained in the binary logistic regressions. Otherwise, the results may be unstable.

Reviewer 2 Report

The article is very interesting and deals with an important topic. The problem of aging populations affects not only China, but also most societies in highly developed countries. Therefore, therapeutic models targeting older women living alone should be sought. 

It is necessary to know the factors associated with poor well-being and loneliness in order to well develop holistic therapeutic models for elderly women. Therefore, I consider this article important and necessary. 

However, in the reviewer's opinion, the discussion should be supplemented. Some examples of multimodal therapeutic interventions for older women should be shown. This will enrich the reader's knowledge of existing solutions. 

I recommend reviewing the publication by Szczepanska-Gieracha and co-authors, as this team of scientists and clinicians has been involved in conducting health promotion programs for older women for many years.

Author Response

Response to Reviewer 2 Comments

Point 1: However, in the reviewer's opinion, the discussion should be supplemented. Some examples of multimodal therapeutic interventions for older women should be shown. This will enrich the reader's knowledge of existing solutions. 

I recommend reviewing the publication by Szczepanska-Gieracha and co-authors, as this team of scientists and clinicians has been involved in conducting health promotion programs for older women for many years.

Response 1: We have added a the discussion using Szczepanska-Gieracha and co-authors' goal-focused group psychotherapy or virtual therapy for older women. Please check Pages 10-11 line 337-353.

Round 2

Reviewer 1 Report

I have no futher comment.